# Curbing gastrointestinal infections by defensin fragment modifications without harming commensal microbiota

Louis Koeninger [1✉], Lisa Osbelt[2,3], Anne Berscheid [4,5], Judith Wendler[1], Jürgen Berger[6], Katharina Hipp [6], Till R. Lesker[2], Marina C. Pils[7], Nisar P. Malek[1], Benjamin A. H. Jensen[8], Heike Brötz-Oesterhelt [4,5,9], Till Strowig [2,10,11] & Jan Wehkamp[1,9,11]

The occurrence and spread of multidrug-resistant pathogens, especially bacteria from the ESKAPE panel, increases the risk to succumb to untreatable infections. We developed a novel antimicrobial peptide, Pam-3, with antibacterial and antibiofilm properties to counter this threat. The peptide is based on an eight-amino acid carboxyl-terminal fragment of human β-defensin 1. Pam-3 exhibited prominent antimicrobial activity against multidrug-resistant ESKAPE pathogens and additionally eradicated already established biofilms in vitro, primarily by disrupting membrane integrity of its target cell. Importantly, prolonged exposure did not result in drug-resistance to Pam-3. In mouse models, Pam-3 selectively reduced acute intestinal *Salmonella* and established *Citrobacter* infections, without compromising the core microbiota, hence displaying an added benefit to traditional broad-spectrum antibiotics. In conclusion, our data support the development of defensin-derived antimicrobial agents as a novel approach to fight multidrug-resistant bacteria, where Pam-3 appears as a particularly promising microbiota-preserving candidate.

---

[1] Department of Internal Medicine I, University Hospital Tübingen, Tübingen, Germany. [2] Department of Microbial Immune Regulation, Helmholtz Centre for Infection Research, Braunschweig, Germany. [3] ESF International Graduate School on Analysis, Imaging and Modelling of Neuronal and Inflammatory Processes, Otto-von-Guericke University, Magdeburg, Germany. [4] Department for Microbial Bioactive Compounds, Interfaculty Institute of Microbiology and Infection Medicine, University of Tübingen, Tübingen, Germany. [5] German Center for Infection Research (DZIF), Partner Site Tübingen, Tübingen, Germany. [6] Max-Planck Institute for Developmental Biology, Electron Microscopy, Tübingen, Germany. [7] Mouse Pathology and Histology, Helmholtz Centre for Infection Research, Braunschweig, Germany. [8] Novo Nordisk Foundation Center for Basic Metabolic Research, Human Genomics and Metagenomics in Metabolism, Faculty of Health and Medical Sciences, University of Copenhagen, Copenhagen, Denmark. [9] Cluster of Excellence - Controlling Microbes to Fight Infections, Tübingen, Germany. [10] Cluster of Excellence - Resolving Infection Susceptibility, Hannover, Germany. [11] These authors contributed equally: Till Strowig, Jan Wehkamp. ✉email: louis.koeninger@med.uni-tuebingen.de

  **1**

The spread of antibiotic-resistant bacteria is an urgent public health threat. Specifically, the so-called ESKAPE (*Enterococcus faecium, Staphylococcus aureus, Klebsiella pneumoniae, Acinetobacter baumannii, Pseudomonas aeruginosa,* and *Enterobacter species*) pathogens account for the majority of nosocomial infections worldwide being attributed to ≥700,000 deaths anually[1]. The World Health Organization (WHO) has recently published a list of 12 bacteria against which new antibiotics are urgently needed, including the ESKAPE pathogens[2]. While traditional antibiotics fight pathogens, they also have wide-ranging consequences for the commensal gut microbiota[3]. Administration disrupts the microbial composition and can result in a long-lasting dysbiosis, which is associated with mounting diseases[4]. Decreased diversity and -taxonomic richness, the spread of antimicrobial resistance as well as increased colonization of opportunistic pathogens, including secondary infections with *Clostridioides difficile*, are just a few of the many side-effects traditional antibiotics impose[5,6]. The current antimicrobial crisis is a product of the long-term neglected development of new antibiotics by pharmaceutical companies and governments[7]. Thus, new strategies more resilient to multidrug resistance are urgently warranted[8].

Antimicrobial peptides (AMPs) are small, cationic peptides existing in all multicellular organisms and exhibit a broad range of antimicrobial and immunological properties[9]. They are considered a promising treatment option and have the potential to be a new generation of antimicrobials against multidrug-resistant bacteria. Recently, de Breij et al. demonstrated the potential of novel antimicrobial peptides by developing SAAP-148, which showed promising effects against biofilm-associated skin infections in ex vivo human skin and murine skin in vivo[10]. First preclinical trials with SAAP-148 against methicillin-resistant *S. aureus* infections have already been conducted according to AdisInsight—a database for drug development[11].

Defensins, the most prominent class of AMPs in humans, are key effector molecules of innate immunity. These peptides protect the host from infectious microbes and shape the composition of microbiota at mucosal surfaces[12–15]. To this end, the first identified human β-defensin, human β-defensin 1 (hBD-1), is constitutively expressed in surface epithelia by monocytes, plasmacyoid dendritic cells, and platelets[16–18]. Previously, the antimicrobial activity of hBD-1 was underestimated until it was analyzed under reduced conditions as found in the human intestine. Reduced hBD-1 has an increased antimicrobial activity, but can be degraded by intestinal proteases[19,20]. We have recently shown that this creates an eight-amino acid carboxyl-terminal fragment (called octapeptide) with retained antimicrobial activity, albeit low in vivo stabilty[21].

Here, we leveraged those findings by developing novel synthetic peptides with improved antimicrobial activity and enhanced in vivo stability. We modified the hBD-1-derived octapeptide with palmitic acid and various spacers, such as sugars or amino acids, to create lipopeptides (Pams) with increased stability and bactericidal activity[22,23]. The most promising peptide was tested against multidrug-resistant pathogens and biofilms followed by exploratory safety assessment. Lastly, we determined its influence on the murine microbiota after oral application as well as its efficacy in murine gastrointestinal infection models.

## Results

**Design and screening of octapeptide based lipopeptides identifies candidate with improved antimicrobial activity.** Lipopeptides are used as antibiotics which are highly active against multidrug-resistant bacteria and fungi[24,25]. Previous studies have demonstrated an enhanced activity of HDPs after fatty acid modification at the N-terminal end[26]. Thereby, $C_{14}$–$C_{18}$ long chain fatty acids have proven to be ideal for this purpose[27,28]. Within this work, we used a $C_{16}$ long fatty acid, namely palmitic acid together with different spacers such as sugars or amino acids to improve stability and bactericidal activity of the carboxyl-terminal fragment of hBD-1. We designed 5 unique lipopeptides,

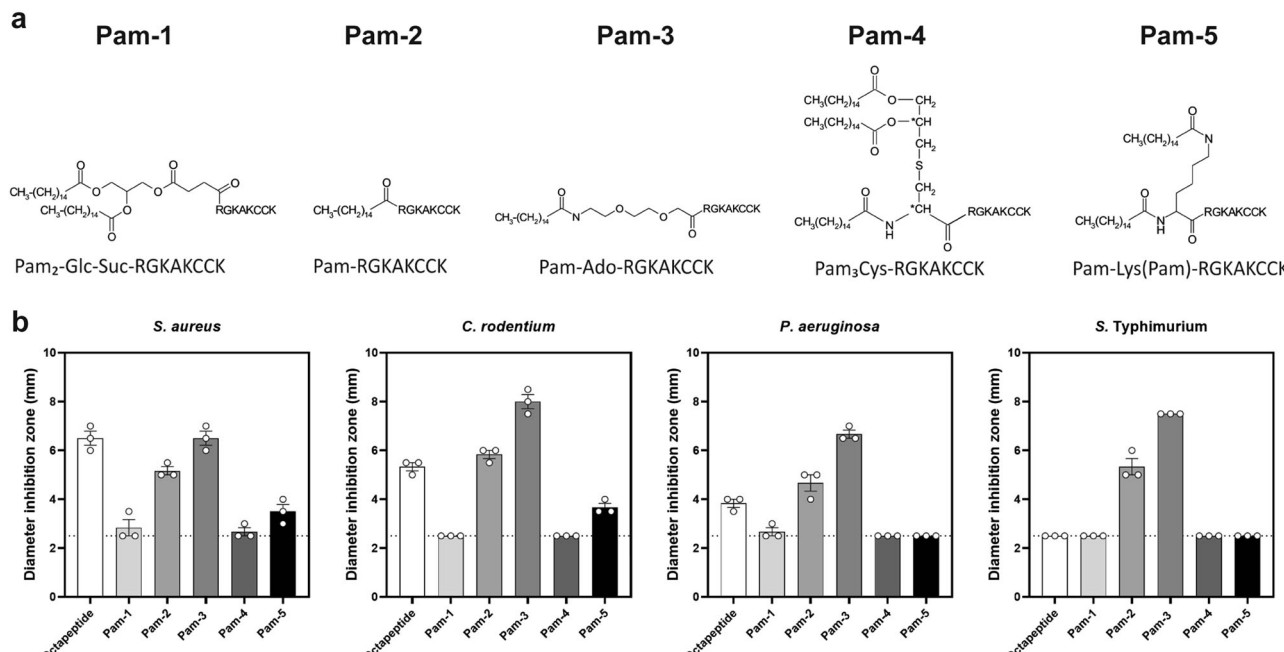

**Fig. 1 Screening of differently modified octapeptides reveals peptides with improved antimicrobial activity. a** Octapeptide, the C-terminal eight amino acids of human β-defensin 1, was chemically modified with palmitic acid and different spacers such as sugars or amino acids or 8-amino-3,6-dioxaoctanoic acid (Ado) to generate lipopeptides. **b** Antimicrobial activity was measured by radial diffusion assay. The diameter of inhibition zones indicates antimicrobial activity; a diameter of 2.5 mm (dotted line) is the diameter of an empty well. Results are means ± SEM of three independent experiments.

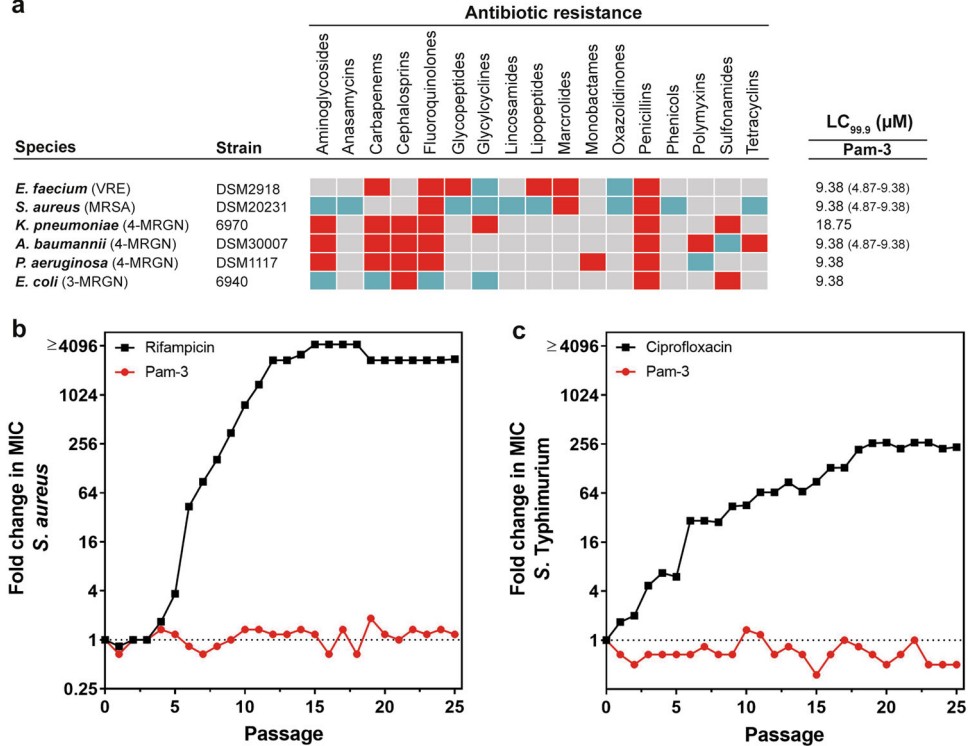

**Fig. 2 Pam-3 kills multidrug-resistant ESKAPE pathogens and resistance was not selected. a** Susceptibility of multidrug-resistant ESKAPE (*E. faecium, S. aureus, K. pneumoniae, P. aeruginosa* and *E. coli*) pathogens to antibiotics and Pam-3. Bacteria susceptible to all (blue boxes) or intermediate/resistant to at least one (red boxes) of the antibiotics per class. Gray boxes are shown if the susceptibility to agents in that class was not assessed. Results are expressed as the $LC_{99.9}$, the lowest peptide concentration in micromolar that resulted in ≥99.9% killing. Results are medians (and ranges) of three independent experiments. If no range is indicated, then the $LC_{99.9}$ was identical in all experiments. **b, c** Resistance development of *S. aureus* ATCC 25923 (**b**) and *S.* Typhimurium DSM554 (**c**) to Pam-3 (red line) and the antibiotics rifampicin and ciprofloxacin (black line), respectively. Values are fold changes (in $\log_2$) in minimal inhibitory concentration (MIC) relative to the MIC of the first passage.

Pam-1 to Pam-5, based on the recently discovered octapeptide (Fig. 1a). This set of peptides was screened for their antimicrobial activity against several pathogenic bacteria using a radial diffusion assay (Fig. 1b). Pam-1, Pam-4, and Pam-5 were generally inactive against tested strains, thus contrasting the potent inhibition of bacterial growth mediated by Pam-2 and Pam-3. Bacterial growth was most strongly inhibited by Pam-3, either on par (*S. aureus*) or superior (*C. rodentium, P. aeruginosa,* and *S.* Typhimurium) to the octapeptide, pointing toward modification-specific activities. Notably, both Pam-2 and Pam-3 consistently inhibited *S.* Typhimurium growth, a species the non-modified octapeptide failed to inhibit.

**Pam-3 displays bactericidal activity against multidrug-resistant bacteria and slow resistance selection.** We used a broth micro-dilution assay to further analyze the potential of our lipopeptides to kill multidrug resistant bacteria belonging to the ESKAPE pathogen panel (Supplementary Fig. 1 and Supplementary Table 1). Pam-1 and Pam-4 displayed no or low bactericidal activity, whereas Pam-5 showed moderate effects against these pathogens. Similar to the results of the radial diffusion assay, both Pam-2 and Pam-3 were highly effective. Remarkably, Pam-2 and Pam-3 inhibited the growth of an *A. baumannii* isolate (DSM30007), which is otherwise resistant to the last-resort antibiotics, colistin and tigecycline. Despite some bactericidal similarities between Pam-2 and Pam-3, the latter proved superior to all other Pam's and was highly effective against these bacteria at concentrations of 4.69–18.75 μM (Fig. 2a and Supplementary Fig. 1). We accordingly selected Pam-3 for further

characterization as a potential therapeutic against multidrug resistant bacteria.

As the development and selection of antibiotic-resistant bacteria in response to new antibiotic candidates is a significant problem, we assessed the ability of *S. aureus* and *S.* Typhimurium to develop resistance against Pam-3. When cultured in the presence of sub-inhibitory concentrations of Pam-3 for 25 passages, no significant increase in the minimal inhibition concentration (MIC) was observed for *S. aureus*. In contrast, the MIC for the standard antibiotic, rifampicin started to rapidly increase after five passages and had increased ≥4096-fold after 15 passages (Fig. 2b). Similarly, although exposed to Pam-3 for continuous serial passages, no resistant *S.* Typhimurium isolates emerged, whereas the presence of ciprofloxacin resulted in an increased MIC already after 3 passages, and a ≥256-fold MIC increase after 19 passages (Fig. 2c).

**Pam-3 eliminates established biofilms and causes rapid killing by permeabilizing the bacterial membrane.** Bacterial biofilms are highly resistant to growth inhibitors and bactericidal treatment regimens. Apart from the hindered penetration of anti-bacterial agents, treatment is further complicated by 10–1000 times increased tolerance exhibited by biofilm protected bacteria compared to planktonic bacteria[29]. Because of that, we assessed the ability of Pam-3 to eradicate established biofilms in a dose-depended manner. Within 1 h, 300 μM of Pam-3 eliminated *P. aeruginosa* in biofilms (Kruskal–Wallis test, $p = 0.0074$, Fig. 3a) and similarly eradicated ~99.99% of *S. aureus* in biofilms (Kruskal–Wallis test, $p = 0.0026$, Fig. 3a). A primary target of antimicrobial peptides is the bacterial cell envelope. Disturbing

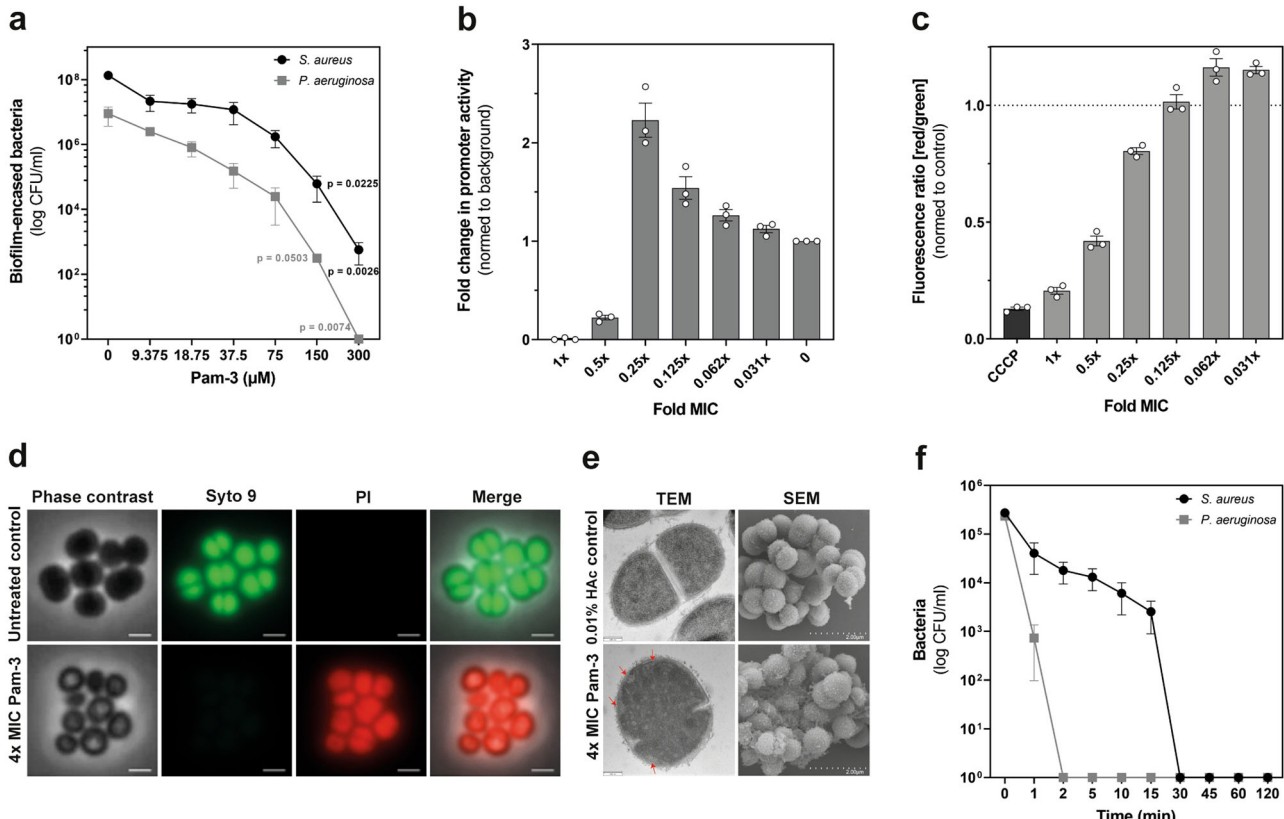

**Fig. 3 Pam-3 eliminates established biofilms, causing rapid killing by inducing large pores or lesions. a** Bactericidal activity of Pam-3 against established biofilms of *S. aureus* ATCC25923 (black line) and *P. aeruginosa* PAO1 (gray line). Results are expressed as the number of viable bacteria (in $\log_{10}$ CFU) after 1 h treatment of 24 h old biofilms with Pam-3. Values are means ± SEM of three replicates from three independent experiments. Statistics were evaluated by using the Kruskal–Wallis test. **b** Activation of the *B. subtilis* ypuA promoter by Pam-3 indicates cell envelope stress. Data are presented as mean ± SEM. Experiments were carried out three independent times. **c** Effect of Pam-3 on *S. aureus* NCTC8325 membrane potential after 30 min of treatment. The protonophore CCCP was used as a positive control and 0.01% acetic acid as a negative control. Data are presented as mean ± SEM of three independent experiments. **d** Fluorescence microscopy of *S. aureus* NCTC8325 treated with Pam-3 (4x MIC) reveals pore formation by causing a strong influx of red-fluorescent propidium iodide. Scale bars: 1 µm. **e** Transmission electron microscopy of high-pressure frozen, freeze-substituted, and embedded *S. aureus* NCTC8325 treated with Pam-3 or 0.01% acetic acid for 30 min to observe membrane disruption. Scale bars, 0.2 µm. *S. aureus* NCTC8325 was exposed to Pam-3 or 0.01% acetic acid for 60 min. The samples were fixed in Karnovsky's reagent, and morphology was analyzed by scanning electron microscopy. Scale bars, 2 µm. **f** Killing of *S. aureus* ATCC25923 (black line) and *P. aeruginosa* PAO1 (gray line) after 1–120 min exposure to 9.38 µM (1x MIC) Pam-3. Results are expressed as the number of viable bacteria (in $\log_{10}$ CFU) per milliliter. Values are means ± SEM of three independent experiments.

the integrity and function of the outer and/or inner membranes results in loss of the barrier function and dissipation of the membrane potential[30]. To clarify the mode of action of Pam-3, we used a *ypuA* promotor-based luciferase reporter strain of *B. subtilis* to identify cytoplasmic membrane-associated and cell envelope-related stress[31]. The *ypuA* promoter was activated (2 fold) by Pam-3, indicating cell envelope impairment (Fig. 3b). Hence, to strengthen this result, we analyzed the influence of Pam-3 on the transmembrane potential of *S. aureus* NCTC8325. The protonophore carbonyl cyanide m-chlorophenyl hydrazone (CCCP) was used as a positive control to depolarize bacteria, i.e., leading to a reduction of their membrane potential. Upon depolarization, $DiOC_2(3)$ shifts from green fluorescence toward red emission because of self-association of the dye molecules. Pam-3 treatment caused a breakdown of the membrane potential in a concentration-dependent manner (Fig. 3c). Both results emphasize that Pam-3 acts on bacterial membranes. Since a large variety of AMPs target the membrane as pore formers[32], we next analyzed the ability of Pam-3 to induce membrane lesions. To this end, we treated *S. aureus* NCTC8325 with Pam-3 at 4x MIC and added a mixture of Syto9 and propidium iodide (PI). The membrane-permeant Syto9 stains all living cells green, whereas

the red-fluorescent PI can only enter cells through large membrane pores or lesions. Pam-3 led to a strong influx of PI (Fig. 3d). To further assess the mechanism of killing, transmission electron microscopy (TEM) was performed to analyze changes in bacterial morphology to compare bacterial morphology before and after treatment with Pam-3. In agreement with the pore formation and the induction of cell envelope stress and depolarization, Pam-3 treatment resulted in strong cell envelope damage with disrupted membranes and pores in most of the cells. Furthermore, additional membranous structures could be observed in many cells similar to rhesus macaque θ-defensins[33]. Additionally, scanning electron microscopy (SEM) was employed to observe cell morphological changes after Pam-3 treatment directly. Exposure to Pam-3 resulted in membrane surface disruption and lysed cells similar to hBD1[20], while control cells exhibited a bright and smooth surface (Fig. 3e). Pore formation was associated with fast killing of these bacteria. We assayed bactericidal kinetics to assess the rapidness of Pam-3 mediated killing. Pam-3 killed more than 90% of *P. aeruginosa* within 1 min and *S. aureus* within 15 min (Fig. 3f). Eradication to the level of detection was observed 2 and 30 min after Pam-3 treatment, respectively.

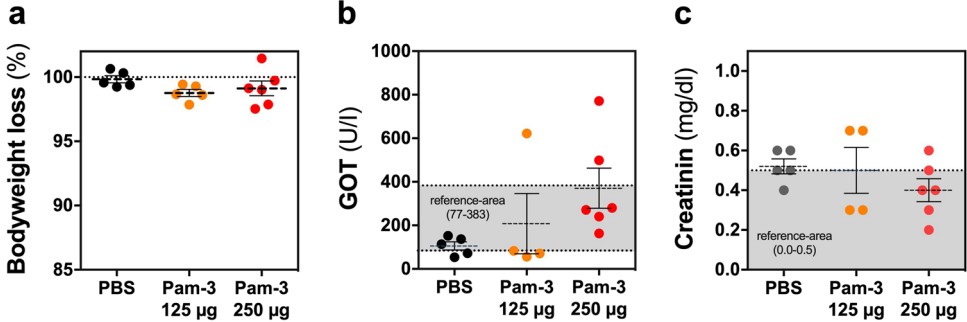

**Fig. 4 Safety of orally applied Pam-3 in mice.** Dose-depended oral tolerance test in mice. Animals were treated twice with 125 μg (orange circles) or 250 μg Pam-3 (red circles) or PBS (black circles). **a** Weight change of mice (125 μg Pam-3, N = 5; 250 μg Pam-3, N = 6 and PBS, N = 5), **b** glutamic oxaloacetic transaminase (GOT) levels of treated animals with 125 μg Pam-3 (N = 4) or 250 μg Pam-3 (N = 6) or PBS (N = 5) and **c** creatinine levels (125 μg Pam-3, N = 4; 250 μg Pam-3, N = 6 and PBS, N = 5) one day after Pam-3 application. Results are presented as mean ± SEM of biologically independent animals.

## Table 1 Safety of orally application of Pam-3 in mice.

**Gastrointestinal tract**

|  | PBS | Pam-3 |
|---|---|---|
| **Stomach** | | |
| Glycogen | 0/5 | 0/6 |
| Lymphocytic aggregates | 0/5 | 0/6 |
| **Duodenum-Jejunum** | | |
| Dysplasia | 0/5 | 0/6 |
| Inflammation | 0/5 | 0/6 |
| **Jejunum-Ileum** | | |
| Dysplasia | 1/5 | 2/6 |
| Inflammation | 0/5 | 0/6 |
| Paneth cells | 0/5 | 0/6 |
| **Cecum and Colon** | | |
| Dysplasia | 0/5 | 0/6 |
| Inflammation | 0/5 | 0/6 |
| **Liver** | | |
| Glycogen | 0/5 | 0/6 |
| Lymphocytic aggregates | 1/5 | 0/6 |
| Anisocaryosis | 0/5 | 0/6 |
| Double nucleated cells | 0/5 | 0/6 |
| Hematopoiesis | 0/5 | 0/6 |
| Fatty change | 0/5 | 0/6 |
| **Kidney** | | |
| Glomeruli | 0/4 | 0/6 |
| Tubuli | 3/4 | 3/6 |
| Papilla | 0/4 | 0/6 |
| Pelvis | 0/4 | 0/6 |

Oral tolerance test in mice. Animals were treated twice with 250 μg Pam-3 or PBS. Results are expressed as the number of the total number of animals within the groups that showed signs of pathology within 24 h after treatment.

**Orally administered Pam-3 showed good acute tolerability in an animal model.** Potential side effects of orally administered Pam-3 were assessed in mice. Histological analysis and determination of serum markers 24 h after application of two doses of 250 μg Pam-3 did not reveal acute toxicity. Specifically, there were no alterations in bodyweight and no signs of systemic toxicity or distress (Fig. 4a). Moreover, measurement of serum levels of glutamic oxaloacetic transaminase and creatinine showed no significant differences between the groups suggesting no effect on kidney and liver metabolism (Fig. 4b, c). Finally, histological examination of gastrointestinal tissues, liver, and kidney revealed no alterations, except a minor shortening of intestinal villi the jejunum-ileum of one control and two treated animals (Table 1). As no difference in these histopathological findings was observed between PBS and Pam-3 treated animals, they were regarded as background observations (Table 1 and Supplementary Fig. 2). Thus, we conclude that Pam-3 treatment was not associated with acute toxicity.

**Pam-3 preserves the core gut microbiota.** To investigate the effect of Pam-3 on the intestinal microbiota, we treated mice twice at an 8-h interval with Pam-3 (125 or 250 μg/each dose) or PBS orally and collected fresh fecal samples before and 24 h after application. Microbiota composition was analyzed using 16S rRNA sequencing. Analysis of beta diversity and calculation of weighted Unifrac Distances demonstrated that changes in the microbiota between the before and after samples were similar between Pam-3 treated mice and PBS gavaged control mice (Fig. 5a, b). Similarly, while minor changes in the community structure were observed in both groups (i.e., treated and untreated), the number of detected species as well as the complexity (Wilcoxon-Test, Fig. 5c, d) remained comparable, thus contrasting treatment with traditional antibiotics, such as ampicillin (Supplementary Fig. 3). In line with the analysis of alpha and beta diversity, Pam-3 treatment did not affect the abundance of bacterial genera (Fig. 5e, f). Combined, these results demonstrate that, with the application regime conducted, Pam-3 treatment of healthy chow-fed mice does not affect the overall community structure or diversity of the microbiota.

**Pam-3 treatment combats acute and established gastrointestinal infections of S. Typhimurium and Citrobacter rodentium.** To assess the efficacy of Pam-3 on acute intestinal bacterial infections, mice were infected with S. Typhimurium and treated orally 6 and 22 h post infection with 250 μg peptide or PBS (Fig. 6a). Pam-3 treated animals showed significantly reduced colony-forming units (CFU) of S. Typhimurium in cecum content and tissue (Mann–Whitney test, p < 0.0001 and p = 0.0409, Fig. 6b). Furthermore, Pam-3 also lowered the bacterial load in the small intestine content without affecting the small intestine tissue (Mann–Whitney test, p = 0.0024 and p = 0.8621, Fig. 6c) and tends to reduce weight loss (Fig. 6d).

We subsequently assessed the therapeutic potential of Pam-3 against an already established intestinal infection. To this end, mice were infected with C. rodentium and received two doses of 250 μg Pam-3 or PBS 5 days post infection (Fig. 6a). Pam-3 treatment reduced the number of bacteria in the cecum content and cecum tissue (Mann–Whitney test, p = 0.0104 and p = 0.0473, Fig. 6e). Treatment also significantly reduced the number of viable Citrobacter in colon content and colon tissue (Mann–Whitney test, p = 0.0010 and p = 0.0104, Fig. 6f).

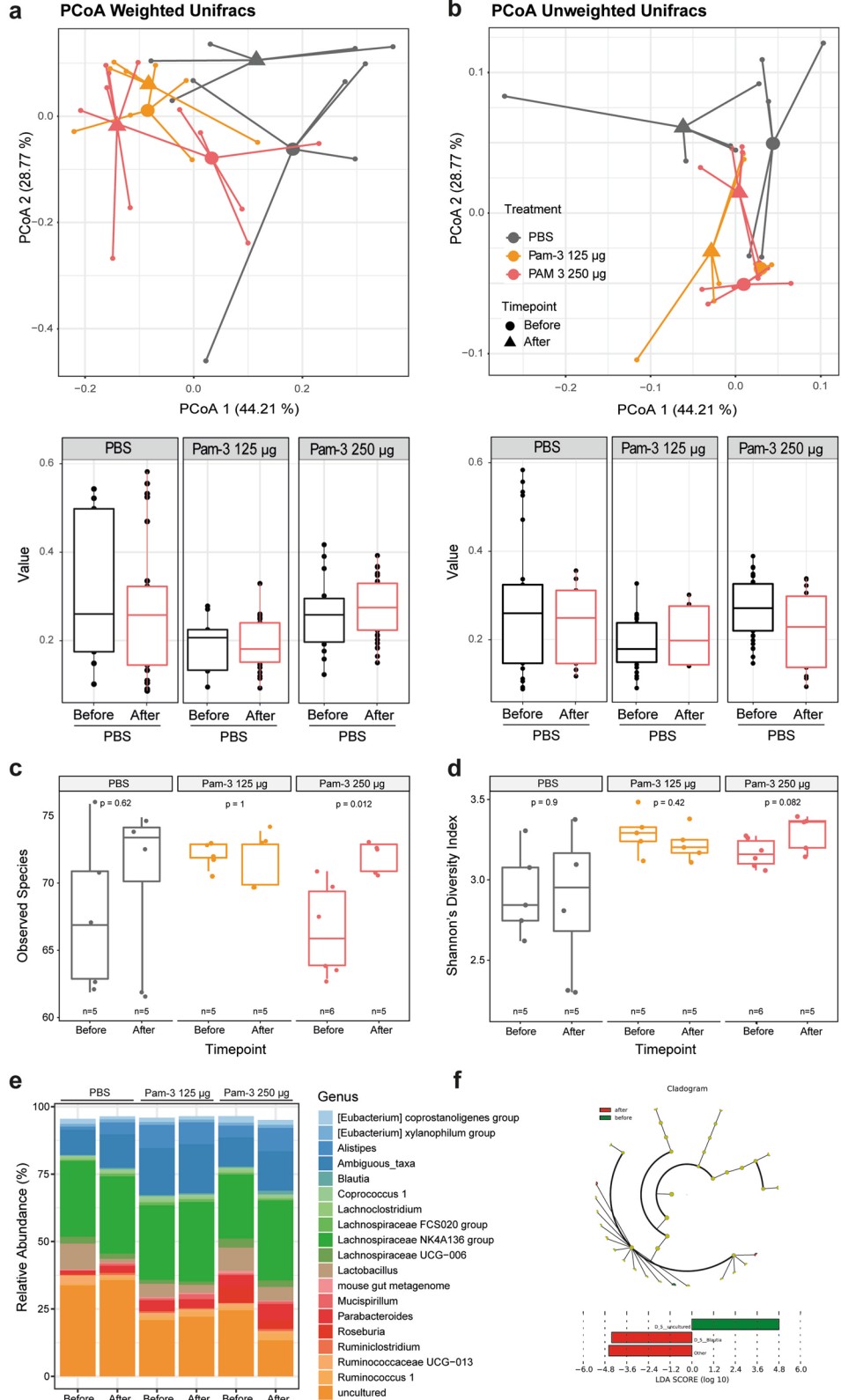

Together these data corroborate the in vivo efficacy of Pam-3 against two different enteric pathogens.

## Discussion

The increasing number of multidrug-resistant pathogens is one of the greatest challenges of our time[34]. The Centers for Disease Control and Prevention (CDC) estimates that the annual infection rate exceeds 2.8 million cases in USA alone, resulting in more than 35,000 deaths caused by multidrug-resistant bacteria and fungi[35]. Due to this alarming development, alternatives to conventional antibiotics are urgently needed[36,37]. Herein, we report that Pam-3, a palmitoleic acid-modified octapeptide fragment from hBD-1, is effective against multidrug-resistant ESKAPE

**Fig. 5 Pam-3 treatment does not affect microbiota diversity but has influence on certain bacteria phyla.** Chow-fed mice were treated orally twice at an 8-h interval with 125 µg Pam-3 ($N = 5$) or 250 µg Pam-3 ($N = 6$) or PBS ($N = 5$) as a control. Feces samples were collected before and after treatment to observe short term changes in the microbiome. **a** Principal coordinate analysis (PCoA including group mean) of fecal microbiota composition using Weighted UniFrac Distances before and after treatment, respectively. **b** PCoA including group mean of fecal microbiota composition using Unweighted UniFrac Distances, respectively before and after treatment. **c** Richness (observed species) before and after treatment. **d** Fecal microbiota was calculated by Shannon's Diversity index. The statistical significance was calculated by using Wilcoxon test. **e** Pam-3 treatment affects the abundance of bacterial genera. **f** Aggregated by genus. Statistical analysis performed by the LEfSe platform (https://galaxyproject.org/learn/visualization/custom/lefse/) using default settings.

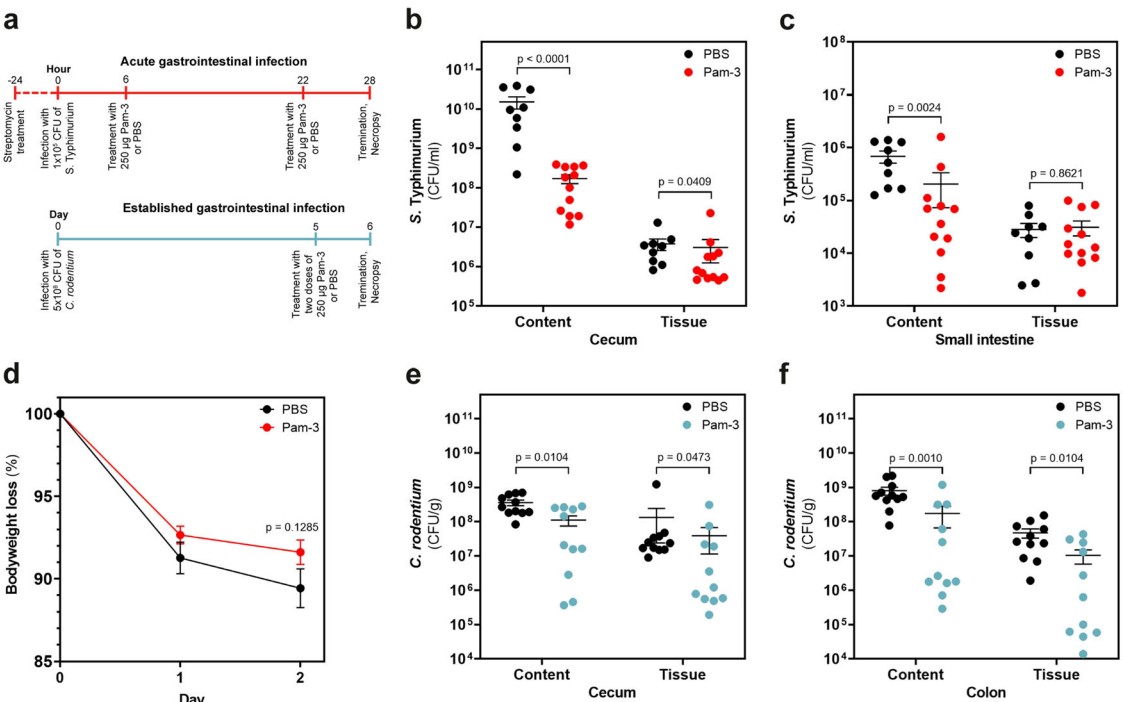

**Fig. 6 Oral application of Pam-3 combats acute and established infections of *S. Typhimurium* and *C. rodentium*. a** Overview of experimental setup. Mice were infected with *S. Typhimurium* (red circles) and treated orally with either 250 µg Pam-3 ($N = 12$) or PBS ($N = 9$) after 6 h and 22 h post infection. Mice were infected with *C. rodentium* (blue circles) and treated orally with either 250 µg Pam-3 ($N = 11$) or PBS ($N = 11$) after 5 days post infection. **b** CFU/ml of *S. Typhimurium* in cecum content and tissue. **c** CFU/ml of *S. Typhimurium* in small intestine content and tissue. **d** Body weight change during acute *S. Typhimurium* infection. **e** CFU/g of *C. rodentium* in cecum content and tissue. **f** CFU/g of *C. rodentium* in colon content and tissue. Results are expressed as the number of viable bacteria (in $\log_{10}$ CFU) in the lumen and tissue and presented as mean ± SEM of biologically independent animals. Statistics were evaluated by using the Mann–Whitney test.

pathogens in vitro. Moreover, Pam-3 is also highly effective in vitro against *A. baummanii* resistant to the last-resort antibiotics, colistin and tigecycline. Biofilm-encased bacteria are much less susceptible to conventional antibiotics than their identical planktonic counterparts complicating treatments[38]. In contrast to most antibiotics, our experiments showed that Pam-3 was able to eradicate established *S. aureus* and *P. aeruginosa* biofilms in vitro.

Development of antibiotic resistance is increasing at an alarming rate[39]. Here, we demonstrated the lack of resistance development to Pam-3 in Gram-positive (*S. aureus*) and Gram-negative (*S. Typhimurium*) bacteria compared to conventional antibiotics, when cultured for 25 passages in the presence of subinhibitory concentrations. This result indicates that resistance development against Pam-3 is a rare event[40].

One of the reasons for this observation could be the rapid killing of bacteria by Pam-3 and its associated mode of action. AMPs with bactericidal effects often interact with membranes as part of their mode(s) of action. Apart from membrane-disruptive mechanisms, like pore formation, AMPs can kill through electrostatics and localized perturbations or non-membrane-

disruptive mechanisms, which targets multiple microbial processes and/or physiological functions[41–43].

Our analysis revealed that Pam-3 causes cell envelope stress by breaking down the membrane potential and inducing pore formation. Taken together, the capacity to induce rapid killing of Gram-positive and Gram-negative bacteria, a low risk of resistance selection, combined with biofilm eradication underscores the value of Pam-3 for further drug development.

The rise in antibiotic resistance and the urgently search for potential alternatives shifted greater research focus on AMPs[37,44]. Most AMPs in pre- and clinical development are evaluated for topical rather than oral administration, for different reasons such as in vivo efficacy and stability[45]. We demonstrated the efficiency of orally administered Pam-3 in two different in vivo models of gastrointestinal infections, namely *S. Typhimurium* and *C. rodentium*. In both models, Pam-3 treatment resulted in significantly reduced bacterial burden in the gastrointestinal tract. Further studies, including proper dose regimens, are necessary for full eradication of both infections.

Antibiotic treatment has non negligible consequences. Observational, clinical, and epidemiologic studies have demonstrated

that antibiotic treatment affects the gut microbiota composition with immediate effects on health[5,46]. Changes in the microbiota composition, decreased diversity, reduced taxonomic richness, and a so-called dysbiosis are the main consequences[47,48]. Further, antibiotics can have long-term effects such as increased susceptibility to infections, obesity, and obesity-associated metabolic diseases[5,49]. In contrast to conventional antibiotics, Pam-3 treatment had unexpectedly no appreciable effects on commensal microbes. Although further studies are warranted to rigorously address this assumption, our data points toward pathogen specific killing by Pam-3 treatment, which would be a significant advantage over conventional antibiotics where disruptive effect on the resident microbiota as well as a rapid drop in diversity is commonly observed[50]. Furthermore, after antibiotic treatment, the intestine is often colonized by non-commensal bacteria, which can result in long-term environmental changes[51]. Instead of a loss of diversity, high-dose Pam-3 treated mice showed an unaffected bacterial diversity. Our data points toward an increased number of observed species and nominally increased diversity. This observation is in line with previous studies on other fragments from our group[52] suggesting that targeted attack on certain (rather high-abundant) bacterial strains frees up new niches for low-abundant taxa, which were previously below the limit of detection. Microbiota-modulating capabilities of AMPs could make a difference in the treatment of many gut microbiota associated diseases, including inflammatory bowel disease[53].

In conclusion, the results presented here demonstrate that Pam-3 is a promising alternative to fight multidrug-resistant infections in a post-antibiotic world because of its broad antimicrobial activity against Gram-positive and Gram-negative pathogens and its efficacy against gastrointestinal infections without disrupting the resident microbiota. Future studies are therefore warranted to examine the full potential of this and other biostable novel peptides. Pam-3 could open a new chapter of an effective, microbiota-saving treatment strategy of bacterial infections.

## Methods

**Microorganisms and culture**. Clinical isolates of *A. baumannii* DSM30007, *E. faecium* DSM2918, *K. pneumoniae* DSM30104, and *P. aeruginosa* DSM1117 were provided by the Department for Laboratory Medicine at Robert-Bosch-Hospital Stuttgart, Germany. *C. rodentium* DSM16636 and *E. coli* DSM8695 were obtained from the Deutsche Sammlung von Mikroorganismen und Zellkultur GmbH Braunschweig, Germany. Clinical isolates of *A. baumannii* LMG944, *A. baumannii* ECII, *E. coli* 6940, *E. coli* DSM682, *E. faecium* 11037 CHB, *E. faecium* 20218 CHB, *K. pneumoniae* 6727 and *K. pneumoniae* 6970 as well as *S. aureus* DSM20231, *S. aureus* ATCC25923, *S. aureus* ATCC33592, *S. aureus* ATCC43300, *S. enterica* serovar Typhimurium DSM554, *P. aeruginosa* ATCC27853, *P. aeruginosa* NRZ01677 and *P. aeruginosa* PAO1 were provided by the Institute of Medical Microbiology and Hygiene Tübingen, Germany. *B. subtilis* ypuA and *S. aureus* NCTC8325 was obtained from the Interfaculty Institute for Microbiology and Infection Medicine, Tübingen, Germany. The wild-type *S.* Typhimurium strain SL1344 harboring a chromosomally integrated luxCDABE cassette, which is confirmed by kanamycin resistance[54] and the nalidixic acid and kanamycin-resistant, bioluminescent *C. rodentium* strain ICC180[55] were obtained from Helmholtz Centre for Infection Research Braunschweig, Germany.

All bacteria were stored in cryo vials (Roth) at −80 °C. Before each experiment, inocula from the frozen stocks were grown overnight at 37 °C on LB or Columbia blood agar plates (BD). For experiments, fresh cultures were prepared in tryptic soy broth (BD).

**Peptides**. All lipopeptides were chemically synthesized by EMC Microcollections GmbH (Tübingen, Germany) and purified by precipitation. In detail, peptide synthesis was performed by solid-phase Fmoc/tert-butyl chemistry on Chloro-(2′-chloro)trityl polystyrene resin (Rapp Polymere, Tübingen, Germany) using an automated peptide synthesizer for multiple peptide synthesis (Syro, MultiSynTech, Germany). Side chain-protecting groups of Fmoc-amino acids were 2,2,4,6,7-pentamethyldihydrobenzofuran-5-sulfonyl (Arg), trityl (Cys) and tert-butyloxycarbonyl (Lys). Fmoc-protected amino acids including Fmoc-8-amino-3,6-dioxaoctanoic acid (Fmoc-Ado-OH, IRIS Biotech GmbH, Germany) were coupled twice (double couplings, sevenfold molar excess of amino acids) by in situ

activation using DIC/HOBt and TCTU with DIEA. The removal of the Fmoc-protecting group was carried out twice by treatment with piperidine/DMF (1:4, v/v). Resins were washed with DMF (6x) after each coupling and deprotection step. N-terminal acylation of the resin bound peptide was performed manually using palmitic acid (1.2 eq), DIC/HOBt (1.5 eq) with DIEA (2 eq) in NMP/DCM (4/1) for 16 h. After washing with DMF (8x) and DCM (5x) completeness of acylation was confirmed by Kaiser-Test. The lipopeptide was cleaved off the resin and side-chain deprotected by treatment with TFA/reagent K/H2O (80/15/5, v/v/v) for 3 h and precipitated by adding diethyl ether. After centrifugation, the peptides were dissolved in tert-butyl alcohol/H2O (4:1, v/v) and lyophilized. The structure of the lipopeptide was confirmed by RP-HPLC-ESI-MS and was >>90% (Supplementary Fig. 4). All peptides were dissolved in 0.01% acetic acid.

**Antimicrobial activity**. Antimicrobial activity was analyzed by radial diffusion assay[56]. Log-phase bacteria were cultivated for up to 18 h in TSB (TSB, Becton Dickinson, USA), washed and diluted to $4 \times 10^6$ CFU in 10 ml agar. Bacteria were incubated in 10 ml of 10 mM sodium phosphate, pH 7.4, containing 0.3 mg/ml of TSB powder and 1% (w/v) low EEO-agarose (AppliChem). 1 µg of each lipopeptide was pipetted into punched wells in a final volume of 4 µl and diffused into the gel for 3 h at 37 °C. After that, a nutrient-rich gel with 6% TSB (w/v) and 1% agarose in 10 mM sodium phosphate buffer was poured on top of the first gel and incubated for up to 24 h at 37 °C. Then the diameter of inhibition zones was measured.

**Bactericidal activity**. Bactericidal activity was assessed by broth microdilution assay[57]. Log-phase bacteria were collected by centrifugation (2500 rpm, 10 min, 4 °C), washed twice with 10 mM sodium phosphate buffer containing 1% (w/v) TSB and the optical density at 600 nm was adjusted to 0.1. Approximately $5 \times 10^5$ CFU/ml bacteria were incubated with serial peptide concentrations (1.17–150 µM) in a final volume of 100 µl in 10 mM sodium phosphate buffer containing 1% (w/v) TSB for 2 h at 37 °C. After incubation, 100 µl of 6% TSB (w/v) were added and absorbance was measured at 600 nm (Tecan, Switzerland) and monitored for 18 h. Afterward, 100 µl per well were plated on LB agar to determine the numbers of viable bacteria microbiologically. Bactericidal activity is expressed as the LC$_{99.9}$, the lowest concentration that killed ≥99.9 % of bacteria.

For time-kill experiments, bacteria ($5 \times 10^5$ CFU/ml) were incubated with 9.38 µM Pam-3 in 10 mM sodium phosphate buffer containing 1% (w/v) TSB in LoBind tubes (Eppendorf) in a total volume of 550 µl. As an untreated control, bacteria were incubated in 10 mM sodium phosphate buffer containing 1% (w/v) TSB. After incubation at 37 °C and 150 rpm for 1–120 min, a sample of 50 µl was taken from the suspension and added to 50 µl of a 0.05% (v/v) sodium polyanethol sulfonate (Sigma-Aldrich) solution, which neutralizes remaining peptide activity, and plated on LB agar to determine the number of viable bacteria.

**Resistance development**. Development of resistance to the peptides was assessed with *S. aureus* and *S.* Typhimurium. For comparison, the development of resistance to the clinically relevant antibiotics rifampicin and ciprofloxacin (Sigma-Aldrich) was determined. Bacteria were cultured overnight at 37 °C at 150 rpm in TSB. Bacteria were washed twice with 10 mM sodium phosphate buffer containing 1% (w/v) TSB. Washed bacteria were incubated with serial Pam-3 or antibiotic concentrations (with final concentrations of 1.17–150 µM peptide or 0.0156–0.5 µg/ml rifampicin or ciprofloxacin) in a final volume of 100 µl in 10 mM sodium phosphate buffer containing 1% (w/v) TSB for 2 h at 37 °C. After incubation, 100 µl of 6% TSB (w/v) were added and plates incubated in a humidified atmosphere for 21 h at 37 °C and 150 rpm.

The MIC, the lowest concentration of peptide/antibiotic that caused a lack of visible bacterial growth, was determined for each bacterial species. Thereafter, $5 \times 10^5$ CFU/ml of the 0.5-fold MIC suspension was added to a fresh medium containing peptides/antibiotics and these mixtures were incubated as described above. This was repeated for 25 passages.

**Treatment of established biofilms**. A log-phase culture of *P. aeruginosa* was diluted in BM2 medium and of *S. aureus* in TSB to $5 \times 10^5$ CFU/ml. 100 µl of each bacterial suspension was added to a round-bottom polystyrene microtiter plate and incubated for 24 h at 37 °C in a humidified atmosphere. Then, planktonic bacteria were removed by two wash steps with PBS. Next, biofilms were exposed to serial peptide dilutions (9.38–300 µM) in a final volume of 100 µl in 10 mM sodium phosphate buffer containing 1% (w/v) TSB for 1 h at 37 °C in a humidified atmosphere. As a control, bacteria were exposed to 10 mM sodium phosphate buffer containing 1% (w/v) TSB without peptide. Afterward, adherent bacteria in each well were resuspended, and the number of viable bacteria was determined microbiologically. To visualize the data on a logarithmic scale, a value of 1 CFU was assigned when no growth occurred. The biofilm degradation assay was performed in agreement with the original report describing this method first[58].

**Interaction with the bacterial membrane**. A specific bacterial reporter strain with the genetic background of *Bacillus subitilis* 1S34, carrying the promoter of the ypuA gene, fused to the firefly luciferase reporter gene, was used to identify cell envelope-related damage caused by treatment with antimicrobial compounds[31]. The assay was carried out in agreement with former reports describing this method[20,59].

Bacteria were cultured to an $OD_{600}$ of 0.9 in LB broth with 5 μg/ml erythromycin at 37 °C and diluted to an $OD_{600}$ of 0.02. Serial peptide dilutions (0.15–150 μM) were prepared in a microtiter plate and incubated with the adjusted bacterial suspension at 37 °C for 1 h. Subsequently, citrate buffer (0.1 M, pH 5) containing 2 mM luciferin (Iris Biotech, Germany) was added and luminescence was measured using a microplate reader (Tecan, Switzerland).

**Bacterial membrane potential.** For determination of membrane potential changes, *S. aureus* NCTC8325 was grown to log-phase in LB + 0.1% glucose, harvested and the optical density at 600 nm ($OD_{600}$) was adjusted to 0.5. Bacteria were incubated with 30 μM 3,3′-diethyloxacarbocyanine iodide (DiOC$_2$(3), Invitrogen™) for 15 min in the dark and treated with serial peptide concentrations for 30 min. The protonophore carbonyl cyanide m-chlorophenyl hydrazone (CCCP, Sigma Aldrich) was used as a positive control and DMSO or 0.01% acetic acid as negative controls. Fluorescence was measured at an excitation wavelength of 485 nm and two emission wavelengths, 530 nm (green) and 630 (nm) red, using a microplate reader (Tecan, Switzerland).

**Bacterial pore formation.** Pore formation was monitored using the Live/Dead BacLight bacterial viability kit (Molecular Probes)[60]. *S. aureus* was grown in LB at 37 °C to log-phase, and 100 μl aliquots were treated with 37.5 μM Pam-3 (4x MIC) or left untreated as a control. Samples were taken after 10 min of peptide treatment, then 0.2 μl of a 1:1 mixture of SYTO9 and propidium iodide (PI) was added and further incubated for 15 min at RT in the dark. Fluorescence microscopy was carried out using a Zeiss Axio Observer Z1 automated microscope. Images were acquired with an Orca Flash 4.0 V2 camera (Hamamatsu), C Plan-Apo 63x/1.4 Oil DIC and alpha Plan-Apochromat 100x/1.46 Oil Ph3 objectives (Zeiss) and processed using the Zen software package (Zeiss).

**Transmission and scanning electron microscopy.** Morphologic analysis of bacteria was characterized by electron microscopy[21]. Approximately $1.2 \times 10^9$ CFU/ml bacteria were incubated with 37.5 μM Pam-3 (4x MIC) in 10 mM sodium phosphate buffer containing 1% (w/v) TSB broth for 30 or 120 min at 37 °C. As a control, bacteria were exposed to 0.01% acetic acid. Afterward bacteria were fixed in Karnovsky's reagent.

For transmission electron microscopy (TEM), bacteria were high-pressure frozen (HPF Compact 03, Engineering Office M. Wohlwend GmbH) in capillaries, freeze-substituted (AFS2, Leica Microsystems) with 2% $OsO_4$ and 0.4% uranyl acetate in acetone as substitution medium and embedded in EPON. Ultrathin sections were stained with uranyl acetate and lead citrate and analyzed with a Tecnai Spirit (Thermo Fisher Scientific) operated at 120 kV.

For scanning electron microscopy (SEM), bacteria were washed in PBS and finally fixed with 1% $OsO_4$ on ice for 1 h. Next, samples were prepared on polylysin-coated coverslips, dehydrated in a graduated series to 100% ethanol and critical point dried (Polaron) with $CO_2$. Finally, samples were sputter-coated with a 3 nm thick layer of platinum (Safematic CCu-010) and examined with a Hitachi Regulus 8230 field emission scanning electron microscope (Hitachi) at an accelerating voltage of 5 kV.

**Mice.** C57BL/6N mice were generated and maintained (including breeding and housing) at the animal facilities of the Helmholtz Centre for Infection Research (HZI) under enhanced specific pathogen-free (SPF) conditions[61]. Animals used in the experiments were gender and age matched. Female and male mice with an age of 8–12 weeks were used. Sterilized food and water ad libitum was provided. Mice were kept under a strict 12-h light cycle (lights on at 7:00 am and off at 7:00 pm) and housed in groups of two to six mice per cage. All mice were euthanized by asphyxiation with $CO_2$ and cervical dislocation. All animal experiments were performed in agreement with the local government of Lower Saxony, Germany (approved permission No. 33.19-42502-04-18/2499).

**Safety.** Age- and gender-matched mice received two doses of peptides (0, 125 μg or 250 μg) solved in 100 μl PBS orally per day. Bodyweight and appearance were recorded. The following day, mice were sacrificed and stomach, kidney, spleen, liver, small intestine, cecum, and colon were removed for histological scoring. Around 1 ml of blood was taken from the heart to measure inflammatory markers, including creatinine in the kidney and the enzyme levels of glutamate-oxalacetate-transaminase (GOT) in the liver.

**Infection and treatment of mice**

*Salmonella Typhimurium infection.* For *S.* Typhimurium infections experiments, age- and sex-matched mice between 10 and 14 weeks of age were used. Both-female and male mice were used in experiments. Water and food were withdrawn for 4 h before mice were treated with 20 mg/mouse of streptomycin by oral gavage. Afterward, mice were supplied with water and food ad libitum. 20 h after strep-tomycin treatment, water and food were withdrawn again, 4 h before the mice were orally infected with $10^5$ CFU of *S.* Typhimurium in 200 μl PBS. Drinking water ad libitum was supplied immediately and food 2 h post infection (p.i.). After 6 and 22 h p.i. mice received 250 μg peptide solved in 100 μl PBS or only PBS orally. 48 h after infection, mice were sacrificed, and intestinal organs were removed to assess the bacterial burden in the lumen and tissues. Mice were weighed every day to record potential body-weight loss.

*Citrobacter rodentium infection.* Bioluminescence expressing *C. rodentium* strain ICC180 was used for all infection experiments[55]. *C. rodentium* inocula were prepared by culturing bacteria overnight at 37 °C in LB broth with 50 μg/ml kana-mycin. Subsequently, the culture was diluted 1:100 in fresh medium, and sub-cultured for 4 h at 37 °C in LB broth[62]. Bacteria were washed twice in phosphate-buffered saline (PBS). Mice were orally inoculated with $10^8$ CFU of *C. rodentium* diluted in 200 μl PBS. Weight of the mice was monitored and feces were collected for measurements of the pathogen burden. After 5 days post infection mice received twice 250 μg peptide solved in 100 μl PBS or only PBS orally. The fol-lowing day, mice were sacrificed to assess bacterial burden in the lumen and tissues of the cecum and the colon.

**Analysis of bacterial loads in feces.** Fresh fecal samples were collected and weighted. Samples were homogenized in 1 ml LB media by bead-beating with 1 mm zirconia/silica beads twice for 25 s using a Mini-Beadbeater-96 (BioSpec). To determine CFUs, dilutions of homogenized samples were plated on LB plates with 50 μg/ml kanamycin.

**Analysis of bacterial loads in intestinal content and systemic organs.** All mice were euthanized by asphyxiation with $CO_2$ at indicated time points. Intestinal tis-sues (small intestine, cecum, colon) were removed aseptically. To collect intestinal content, organs were flushed with PBS. Organs were opened longitudinally, cleaned thoroughly with PBS and weighted. Organs and content were homogenized in PBS using a Polytron homogenizer (Kinemtatica). Dilutions of homogenized samples were plated on LB plates containing 50 μg/ml kanamycin to determine CFUs.

**Microbiota analysis.** Feces samples were collected at different time points (before and after infection), and bacterial DNA was extracted from snap-frozen feces using a phenol-chloroform-based method previously described[63]. In brief, 500 μl of extraction buffer (200 mM Tris (Roth), 20 mM EDTA (Roth), 200 mM NaCl (Roth), pH 8.0), 200 μl of 20% SDS (AppliChem), 500 μl of phenol:chloroform: isoamyl alcohol (PCI) (24:24:1) (Roth) and 100 μl of zirconia/silica beads (0.1 mm diameter) (Roth) were added per feces sample. Lysis of bacteria was performed by mechanical disruption using a Mini-BeadBeater-96 (BioSpec) for two times 2 min. After centrifugation, the aqueous phase was processed by another phenol:chloro-form:isoamyl alcohol extraction before precipitation of DNA using 500 μl iso-propanol (J.T. Baker) and 0.1 volume of 3 M sodium acetate (Applichem). Samples were incubated at −20 °C for at least several hours or overnight and centrifuged at 4 °C at maximum speed for 20 min. The resulting DNA pellet was washed, dried using a speed vacuum and resuspended in TE Buffer (Applichem) with 100 μg/ml RNase I (Applichem). Crude DNA was column purified (BioBasic Inc.) to remove PCR inhibitors.

16S rRNA gene amplification of the V4 region (F515/R806) was performed according to an established protocol previously described[64]. Briefly, DNA was normalized to 25 ng/μl and used for sequencing PCR with unique 12-base Golary barcodes incorporated via specific primers (obtained from Sigma). PCR was performed using Q5 polymerase (NewEnglandBiolabs) in triplicates for each sample, using PCR conditions of initial denaturation for 30 s at 98 °C, followed by 25 cycles (10 s at 98 °C, 20 s at 55 °C, and 20 s at 72 °C). After pooling and normalization to 10 nM, PCR amplicons were sequenced on an Illumina MiSeq platform via 250 bp paired-end sequencing (PE250). Using Usearch8.1 software package (http://www.drive5.com/usearch/) the resulting reads were assembled, filtered and clustered. Sequences were filtered for low-quality reads and binned based on sample-specific barcodes using QIIME v1.8.0[65]. Merging was performed using -fastq_mergepairs—with fastq_maxdiffs 30. Quality filtering was conducted with fastq_filter (-fastq_maxee 1), using a minimum read length of 250 bp and a minimum number of reads per sample = 1000. Reads were clustered into 97% ID OTUs by open-reference OTU picking and representative sequences were determined by use of UPARSE algorithm[66]. Abundance filtering (OTUs cluster >0.5%) and taxonomic classification were performed using the RDP Classifier executed at 80% bootstrap confidence cut off[67]. Sequences without matching reference dataset were assembled as de novo using UCLUST. Phylogenetic relationships between OTUs were determined using FastTree to the PyNAST alignment[68]. Resulting OTU absolute abundance table and mapping file were used for statistical analyses and data visualization in the R statistical programming environment package phyloseq[69].

**Statistics and reproducibility.** Experimental results were analyzed for statistical significance using GraphPad Prism v8.2 (GraphPad Software Inc.). Apart from microbiota analysis, all data were analyzed by Mann–Whitney test or Kruskal–Wallis test. Here, differences were analyzed by Wilcoxon signed-rank test or Kruskal–Wallis test comparison of totals between groups[70]. OTUs with Kruskal–Wallis test <0.05 were considered for analysis.

**Reporting summary**. Further information on research design is available in the Nature Research Reporting Summary linked to this article.

## Data availability

16S rRNA gene sequencing data have been deposited in the NCBI (Bioproject Database) under the accession number: PRJEB37278. All source data underlying the graphs presented in the main figures are available in Supplementary Data 1. Please contact the corresponding author (L.K.) for access of other data or information, which is stored electronically, and will be made available upon reasonable requests.

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

## Acknowledgements

We thank Marion Strauss and Jutta Bader for excellent technical assistance; Stefanie Fehrenbacher for providing the Antibiograms of ESKAPE pathogens. This study was supported by the Deutsche Forschungsgemeinschaft (DFG, German Research Foundation) to Jan Wehkamp—Project ID WE4336/7-1 and also under Germany´s Excellence Strategy—EXC 2124 "CMFI"—Project ID 390838134. Till Strowig was supported by the Helmholtz Association and the DFG under Germany's Excellence Strategy—EXC 2155 "RESIST"—Project ID 39087428. Heike Brötz-Oesterhelt and Anne Berscheid acknowledge funding by the German Center of Infection Research (DZIF), project TTU 09.818, and infrastructural support by the Deutsche Forschungsgemeinschaft (DFG, German Research Foundation) under Germany´s Excellence Strategy—EXC 2124 "CMFI"—Project ID 390838134. Lisa Osbelt was supported by the federal state of Saxony-Anhalt and the European Structural and Investment Funds (ESF, 2014–2020), project number 4410032030ZS/2016/08/80645. We acknowledge support by Open Access Publishing Fund of University of Tübingen.

## Author contributions

L.K conceived the research, designed and carried out experiments and data analysis, and wrote the manuscript. L.O. designed, conducted the in vivo infection model experiments and analyzed the data. M.P. conducted histology and analyzed the data. A.B. performed in vitro MoA experiments and data analysis. J.B. and K.H. were in charge for the electron microscopy experiments. T.L. performed the bioinformatics. A.B., J.u.W., N.P.M., B.A.H. J., and H.B.-O. assisted with data interpretation and manuscript editing. T.S. and J.W. conceived the research, assisted with data interpretation and wrote the manuscript. All authors were involved in data discussion and the final version of the manuscript.

## Competing interests

J. Wehkamp and L. Koeninger are planning to apply for a patent on lipopeptides of the "Pam" series. The other authors declare no competing interests.
