## [Peer Review File · Communications Biology]

Reviewers' comments:

Reviewer #1 (Remarks to the Author):

Koeninger, et al report an investigation on an interesting modified octapeptide based on the C-terminal sequence of human beta-defensin 1 (DEFB1). Members of this investigative team previously identified the octapeptide RGKAKCCK as an active domain of DEFB1, but lacked in vivo stability. Here, the authors modified the peptide with palmitic acid (PA) on the N-terminus, directly or using four different spacing linkers. The modified peptides were screened for antimicrobial activity and a PA-modified peptide with an 8-amino-3,6-dioxaoctanoic acid spacer (named "PAM-3") had highest activity. PAM-3 had activity against a collection of multi-resistant ESCAPE pathogens, which are identified by the WHO as high-priority targets for drug development. Repeat exposure (25-cycles of passage) to PAM-3 did not generate resistance in tested strains of *S. aureus* or *S. enterica* serovar Typhimurium (*S. Typhimurium*). PAM-3 also showed bactericidal activity against established biofilms of *S. aureus* and *P. aeruginosa*. Finally, In vivo experiments using a mouse model provide evidence that orally administered PAM-3 is tolerated without overt toxicity, had little effects on the endogenous microbiota and is effective in two models of bacterial enteritis. Together the investigation has identified a promising new candidate antimicrobial agent for drug development.

Major comments

I suggest re-writing the last sentence of the abstract to be more circumspect. "Cost-effective" is not addressed in this paper, and "minimal-side-effect" is not rigorously established.

Similarly, I suggest revising the title. As phrased, it seems to imply a report of in vivo experiments with ESCAPE pathogens, given the phrase "without harming commensal microbiota."

Minor suggestions:

1) Lines 66/67. Add ref #10 to the sentence with de Breij et al. Are preclinical trials reported in reference 10, as implicated on line 70?

2) Line 75. I do not think that reference #15 reports the cellular sources as indicated.

3) Line 148. Consider, "We assayed bactericidal kinetics to assess the rapidness..."

4) Lines 161, 162 and 237. Consider "administered" rather than applied.

5) Lines 213, 214, and 217. Add "in vitro" for clarity.

6) Line240. Consider, "...including proper dose regimens are..."

7) Line 250. "precision editing" is suggested, but not rigorously tested. "...where only pathogenic bacteria are targeted and beneficial commensal bacteria remain unaffected..." is likely an overstatement based on the limited experiments conducted.

8) Line 679. "...the C-terminal eight amino acids..." rather than "...the last eight amino acids..."

9) Figure 2a and line 109. Three decimal places likely over-states the precision of these LC values.

10) Line 726. How many mice per group?

Reviewer #2 (Remarks to the Author):

The authors have reported interesting results with a new synthetic peptide (Pam-3) which could be a promising alternative to fight multidrug resistant infections. The results showed its antimicrobial activity against Gram-positive and Gram-negative pathogens and efficacy against gastrointestinal infections without disrupting the resident microbiota. The experimental tests were properly conducted in vitro and in vivo and the results are well described and supported.

The results are of great importance for combating antimicrobial resistance, opening new possibilities of treatment for this disease. Probably, this article will be cited a lot and may encourage new and important works like this. I believe that the statistical analysis is adequate. The paper is well founded. However, I would like to obtain some clarifications and, in addition, request some additions to the work.

Minor revision

1) Add in the article how the rational design of the Pam-1 to Pam-5 was carried out?

2) Insert the results obtained (in vitro) with the peptides (Pam-1 to Pam-5) in *Citrobacter rodentium* - in Figure 1.

Explain why it was chosen for the oral essay.

3) Likewise, it would be very useful to add the effects of the new peptides (Pam-1 to Pam-5) on the other ESKAPE bacteria, thus showing their possible broad spectrum.

Dear Dr. Calvin Henard, Editorial Board Member of Communications Biology:

We would like to express our sincere gratitude to both you as editor and to the reviewers for constructive comments of our manuscript COMMSBIO-20-0938-T (revised title) ‘Curbing gastrointestinal infections by defensin fragment modifications without harming commensal microbiota’ which we believe has helped us improve the quality of the enclosed MS. To this end, we have carefully revised the MS in accordance with the reviewers’ suggestion. We hope that the revised version meets your expectation and that you will find it suitable for publication in *Communications Biology*. Please find our point-by-point response below and also track our changes within the MS highlighted in yellow.

Sincerely yours,

Louis Koeninger.

Reviewer comments:

Reviewer #1 (Remarks to the author)

I suggest re-writing the last sentence of the abstract to be more circumspect. "Cost-effective" is not addressed in this paper, and "minimal-side-effect" is not rigorously established.

Response: We appreciate this comment and accordingly removed ‘cost-effective and minimal-side-effect’.

Action: Instead we wrote ‘microbiota-preserving candidate’ (Line 44).

Similarly, I suggest revising the title. As phrased, it seems to imply a report of in vivo experiments with ESCAPE pathogens, given the phrase "without harming commensal microbiota."

Response: We acknowledge this suggestion and therefore rephrased the title to ‘Curbing gastrointestinal infections by defensin fragment modifications without harming commensal microbiota’.

Lines 66/67: Add ref #10 to the sentence with de Breij et al. Are preclinical trials reported in reference 10, as implicated on line 70?

Response: We thank the reviewer for pointing this out. Reference 10 was added as suggested by the reviewer. Yes, preclinical trials were scheduled for the first quarter of 2018 in reference 10. To-date, preclinical trials were conducted in various setups according to the drug development and research database AdisInsight.

Action: We have now included this information (Line 69-70).

Line 75: I do not think that reference #15 reports the cellular sources as indicated.

Response: We apologize for this inaccuracy and have now added more appropriate references.

Line 148: Consider, "We assayed bactericidal kinetics to assess the rapidness..."

Response: We thank the reviewer for this suggestion and have now changed line 157 (previously 148) to 'We assayed bactericidal kinetics to assess the rapidness...' as suggested.

Lines 161, 162 and 237: Consider "administered" rather than applied.

Response: Has been corrected to 'administered'.

Lines 213, 214, and 217: Add "in vitro" for clarity.

Response: We agree with this observation and have therefore added "in vitro" for clarity in lines 223, 224 and 228 (previously 213, 214 and 217).

Line 240: Consider, "...including proper dose regimens are..."

Response: Has been corrected to 'including proper dose regimens are'.

Line 250: "precision editing" is suggested, but not rigorously tested. "...where only pathogenic bacteria are targeted and beneficial commensal bacteria remain unaffected..." is likely an overstatement based on the limited experiments conducted.

Response: We appreciate this point and have accordingly chosen a more cautious phrasing in the revised MS (Line 260-263).

Action: Has been corrected to 'Although further studies are warranted to rigorously address this assumption, our data points towards pathogen specific killing by PAM-3 treatment, which would be a significant advantage over conventional antibiotics where disruptive effect on the resident microbiota as well as a rapid drop in diversity is commonly observed⁵⁰.

Line 679: "...the C-terminal eight amino acids..." rather than "...the last eight amino acids..."

Response: Has been corrected to ‘the C-terminal eight amino acids’.

Figure 2a and line 109: Three decimals places likely over-states the precision of these LC values.

Response: We thank the reviewer for this critical point and have conducted additional experiments with various bacteria from the ESKAPE panel. In total, we tested Pam-3 against 18 different bacteria and performed every experiment three times. Pam-3 was highly effective at concentrations of 4.69 to 18.75 μM . The overall measured mean is a concentration of 9.38 μM Pam-3.

Action: We have now included a new extended data figure 1 with these results and provide a more detailed discussion of these new findings (Line 112-120).

Line 726: How many mice per group?

Response: We sincerely apologize for not having included this crucial information in the first version of the manuscript. Five mice were used in the 125 μg Pam-3 and PBS groups, while the 250 μg Pam-3 group consisted of six mice.

Action: We have now included the animal number in line 802-803.

Reviewer #2 (Remarks to the author)

Add in the article how the rational design of the Pam-1 to Pam-5 was carried out?

Response: We thank the reviewer for this critical point and regret the lack of clarity in the first version of the manuscript. We have now specified the rational design of the Pam's in the manuscript (Line 93-98).

Insert the results obtained (in vitro) with the peptides (Pam-1 to Pam-5) in Citrobacter rodentium - in Figure 1. Explain why it was chosen for the oral essay.

Response: We thank the reviewer for his important comment and have conducted a radial diffusion assay with *Citrobacter rodentium*. The result underlines our previous findings that Pam-3 is the most active peptide of this family.

We chose the mouse-restricted pathogen *C. rodentium* to mimic human enterohemorrhagic *E. coli* (EHEC) and enteropathogenic *E. coli* (EPEC) infections, since these bacteria only induce modest pathogenicity in mice. EHEC and EPEC as well as *Salmonella* can cause acute gastroenteritis including severe inflammation and diarrhea. Diarrhea is a global problem with an infection rate of more than 100 million cases per year, responsible for more than 1.6 million deaths. Mostly children in developing countries suffer because of it, resulting in high rates of morbidity and mortality among children younger than 5 years. Still, *Salmonella* outbreaks remain a major problem in hospitals and nursing homes in developed countries. We thus strive to make a contribution to the fight against these pathogens. Most AMPs in pre- and clinical development are evaluated for topical rather than oral administration, but based on our experience from previous studies with orally administered peptides and the above mentioned problem we have chosen the oral application for the Pam's.

Action: We have now included the result for *C. rodentium* in Figure 1b and updated the results section (Line 104).

Likewise, it would be very useful to add the effects of the new peptides (Pam-1 to Pam-5) on the other ESKAPE bacteria, thus showing their possible broad spectrum.

Response: We thank the reviewer for this comment. We performed additional experiments with the Pam's against various bacteria from the ESKAPE panel, now presented in the new extended data figure 1.

Action: We added a new result section (Line 112-120) to address this comment.

Reviewers' comments:

Reviewer #2 (Remarks to the Author):

Dear,

All reported problems have been answered properly.

I believe that article can be published in Communications Biology.

Point-by-point response to the referees' comments

Reviewer comments:

Reviewer #1 (Remarks to the author)

I suggest re-writing the last sentence of the abstract to be more circumspect. "Cost-effective" is not addressed in this paper, and "minimal-side-effect" is not rigorously established.

Response: We appreciate this comment and accordingly removed 'cost-effective and minimal-side-effect'.

Action: Instead we wrote 'microbiota-preserving candidate' (Line 44).

Similarly, I suggest revising the title. As phrased, it seems to imply a report of in vivo experiments with ESCAPE pathogens, given the phrase "without harming commensal microbiota."

Response: We acknowledge this suggestion and therefore rephrased the title to 'Curbing gastrointestinal infections by defensin fragment modifications without harming commensal microbiota'.

Lines 66/67: Add ref #10 to the sentence with de Breij et al. Are preclinical trials reported in reference 10, as implicated on line 70?

Response: We thank the reviewer for pointing this out. Reference 10 was added as suggested by the reviewer. Yes, preclinical trials were scheduled for the first quarter of 2018 in reference 10. To-date, preclinical trials were conducted in various setups according to the drug development and research database AdisInsight.

Action: We have now included this information (Line 69-70).

Line 75: I do not think that reference #15 reports the cellular sources as indicated.

Response: We apologies for this inaccuracy and have now added more appropriate references.

Line 148: Consider, "We assayed bactericidal kinetics to assess the rapidness..."

Response: We thank the reviewer for this suggestion and have now changed line 157 (previously 148) to 'We assayed bactericidal kinetics to assess the rapidness...' as suggested.

Lines 161, 162 and 237: Consider "administered" rather than applied.

Response: Has been corrected to ‘administered’.

Lines 213, 214, and 217: Add "in vitro" for clarity.

Response: We agree with this observation and have therefore added “*in vitro*” for clarity in lines 223, 224 and 228 (previously 213, 214 and 217).

Line 240: Consider, "...including proper dose regimens are..."

Response: Has been corrected to ‘including proper dose regimens are’.

Line 250: "precision editing" is suggested, but not rigorously tested. "...where only pathogenic bacteria are targeted and beneficial commensal bacteria remain unaffected..." is likely an overstatement based on the limited experiments conducted.

Response: We appreciate this point and have accordingly chosen a more cautious phrasing in the revised MS (Line 260-263).

Action: Has been corrected to ‘Although further studies are warranted to rigorously address this assumption, our data points towards pathogen specific killing by PAM-3 treatment, which would be a significant advantage over conventional antibiotics where disruptive effect on the resident microbiota as well as a rapid drop in diversity is commonly observed⁵⁰.

Line 679: "...the C-terminal eight amino acids..." rather than "...the last eight amino acids..."

Response: Has been corrected to ‘the C-terminal eight amino acids’.

Figure 2a and line 109: Three decimals places likely over-states the precision of these LC values.

Response: We thank the reviewer for this critical point and have conducted additional experiments with various bacteria from the ESKAPE panel. In total, we tested Pam-3 against 18 different bacteria and performed every experiment three times. Pam-3 was highly effective at concentrations of 4.69 to 18.75 μM . The overall measured mean is a concentration of 9.38 μM Pam-3.

Action: We have now included a new extended data figure 1 with these results and provide a more detailed discussion of these new findings (Line 112-120).

Line 726: How many mice per group?

Response: We sincerely apologies for not having included this crucial information in the first version of the manuscript. Five mice were used in the 125 μ g Pam-3 and PBS groups, while the 250 μ g Pam-3 group consisted of six mice.

Action: We have now included the animal number in line 802-803.

Reviewer #2 (Remarks to the author)

Add in the article how the rational design of the Pam-1 to Pam-5 was carried out?

Response: We thank the reviewer for this critical point and regret the lack of clarity in the first version of the manuscript. We have now specified the rational design of the Pam's in the manuscript (Line 93-98).

Insert the results obtained (in vitro) with the peptides (Pam-1 to Pam-5) in Citrobacter rodentium - in Figure 1. Explain why it was chosen for the oral essay.

Response: We thank the reviewer for his important comment and have conducted a radial diffusion assay with *Citrobacter rodentium*. The result underlines our previous findings that Pam-3 is the most active peptide of this family.

We chose the mouse-restricted pathogen *C. rodentium* to mimic human enterohemorrhagic *E. coli* (EHEC) and enteropathogenic *E. coli* (EPEC) infections, since these bacteria only induce modest pathogenicity in mice. EHEC and EPEC as well as *Salmonella* can cause acute gastroenteritis including severe inflammation and diarrhea. Diarrhea is a global problem with an infection rate of more than 100 million cases per year, responsible for more than 1.6 million deaths. Mostly children in developing countries suffer because of it, resulting in high rates of morbidity and mortality among children younger than 5 years. Still, *Salmonella* outbreaks remain a major problem in hospitals and nursing homes in developed countries. We thus strive to make a contribution to the fight against these pathogens. Most AMPs in pre- and clinical development are evaluated for topical rather than oral administration, but based on our experience from previous studies with orally administered peptides and the above mentioned problem we have chosen the oral application for the Pam's.

Action: We have now included the result for *C. rodentium* in Figure 1b and updated the results section (Line 104).

Likewise, it would be very useful to add the effects of the new peptides (Pam-1 to Pam-5) on the other ESKAPE bacteria, thus showing their possible broad spectrum.

Response: We thank the reviewer for this comment. We performed additional experiments with the Pam's against various bacteria from the ESKAPE panel, now presented in the new extended data figure 1.

Action: We added a new result section (Line 112-120) to address this comment.